# HKDSME: Heterogeneous Knowledge Distillation for Semi-supervised Singing Melody Extraction Using Harmonic Supervision

## ABSTRACT

Singing melody extraction is a key task in the field of music information retrieval (MIR). However, decades of research works have uncovered two difficult issues. *First*, binary classification on frequency-domain audio features (e.g., spectrogram) is regarded as the primary method, which ignores the potential associations of musical information at different frequency bins, as well as their varying significance for output decisions. *Second*, the existing semi-supervised singing melody extraction models ignore the accuracy of the generated pseudo labels by semi-supervised models, which largely limits the further improvements of the model. To solve the two issues, in this paper, we propose a heterogeneous knowledge distillation framework for semi-supervised singing melody extraction using harmonic supervision, termed as *HKDSME*. We begin by proposing a four-class classification paradigm for determining the results of singing melody extraction using harmonic supervision. This enables the model to capture more information regarding melodic relations in spectrograms. To improve the accuracy issue of pseudo labels, we then build a semi-supervised method by leveraging the extracted harmonics as a consistent regularization. Different from previous methods, it judges the availability of unlabeled data in terms of the inner positional relations of extracted harmonics. To further build a light-weight semi-supervised model, we propose a heterogeneous knowledge distillation (HKD) module, which enables the prior knowledge transfers between heterogeneous models. We also propose a novel confidence guided loss, which incorporates with the proposed HKD module to reduce the wrong pseudo labels. We evaluate our proposed method using several well-known public available datasets, and the findings demonstrate the efficacy of our proposed method.

## CCS CONCEPTS

• **Applied computing** → *Sound and music computing*.

## KEYWORDS

Heterogeneous Knowledge Distillation, Harmonic Supervision, Singing Melody Extraction, Music Information Retrieval

**Unpublished working draft. Not for distribution.**

## 1 INTRODUCTION

Singing melody extraction is a challenging task in the field of music information retrieval (MIR). It aims to extract the fundamental frequency (f0) contour from polyphonic music. Recently it has become an active research topic with a lot of downstream applications, such as cover song identification [40, 53], query-by-humming [44], voice separation [23], and music recommendation [27]. Singing melody contour obtained from extraction models can be utilized as an audio feature of musical information to enhance the performance of these downstream tasks.

With the trend of artificial intelligence, deep learning models play an important role in the development of singing melody extraction techniques. A number of deep learning based methods [3, 21, 42, 49, 50] have been proposed for supervised singing melody extraction. Then, in an attempt to solve the problem of data insufficiency, semi-supervised singing melody extraction methods have become a cutting-edge direction. Some pioneer works [29, 48] adopt semi-supervised learning to utilize unlabeled music tracks as training data for the singing melody extraction task, seeking to improve the melody extraction performance. However, these works either do not contain the process of data selection for unlabeled data or use data augmentation based consistency regularization methods.

Despite these remarkable successes in singing melody extraction, in this paper, we try to further improve the performance of melody extraction models from two perspectives: *a rethinking of supervised training paradigm and an innovation in the approach of semi-supervised singing melody extraction.*

For supervised learning methods of the singing melody extraction task, the prior works employ the binary classification paradigm on all audio pixels in the spectrogram, to predict whether each of them is classified as the f0 or not. However, this paradigm exposes one potential issue in the singing melody extraction task: all pixels in the spectrogram are treated equally as either melodic pixel or non-melodic pixel, while in the actual analysis of music signals, pixels often have different importance. One of the most representative examples is the *harmonic and sub-harmonic information* of the singing melody, which has been proven crucial for singing melody extraction [1, 26, 52]. They are distributed above and below the singing melody line according to a fixed ratio, and synchronize with the changes of the singing melody. If we treat these pixels equally as just melodic or non-melodic pixels, then not only can we not make use of the relationship between these harmonics/sub-harmonics and the singing melody, but they may even in turn be misidentified as the singing melody (known as octave error), because they possess similar patterns and activation values as the singing melody.

For semi-supervised singing melody extraction, while data augmentation is treated as a popular consistency regularization method, the accuracy of pseudo labels is still a challenging issue. Yu et al.

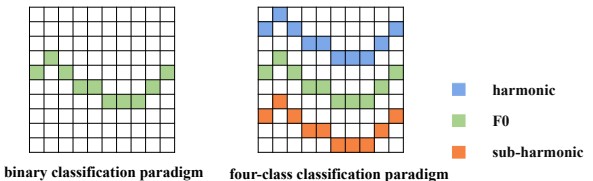

**(a) Binary classification paradigm vs. harmonic supervision paradigm**

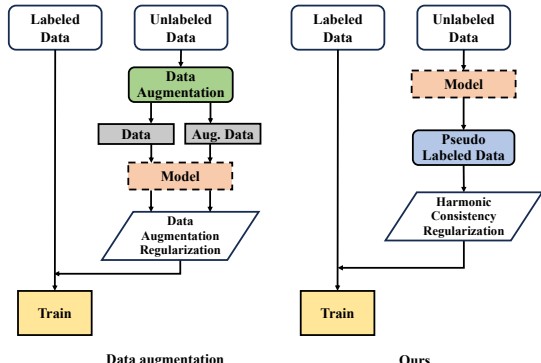

**(b) Data aug. consistency regularization vs. harmonic consistency regularization.**

**Figure 1: Illustration of the proposed harmonic supervision and harmonic consistency regularization.**

[48] has claimed that the singing melody extraction task is very sensitive to the data augmentation, many data augmentation based consistency regularization methods can not obtain satisfied results, such as *MixMatch* [4], *MeanTeacher* [43]. We seek for some method that uses the correlations inside features as a consistency regularization to alleviate this issue. And we expect that such methods can be both light-weight and accurate to further increase the efficiency of the model's utilization of unlabeled data in the semi-supervised learning scenario.

Following the above analysis, in this paper, we first propose a new paradigm, which uses harmonic supervision for supervised singing melody extraction. To be specific, we first leverage harmonic and sub-harmonic information as additional labels, then we perform a four-class classification: *melodic pixels, harmonic pixels, sub-harmonic pixels and non-melodic pixels.* The new paradigm encourages the model to learn the positional correlations in the spectrogram, alleviating the octave error issue.

Extending from supervised learning, we propose to leverage the extracted harmonic and sub-harmonic positional information as a consistency regularization for semi-supervised learning. Different from previous methods, it judges the availability of unlabeled data in terms of the inner positional relations of extracted harmonics. We expect to see how the performance of the task is improved due to the proposed *harmonic consistency regularization* via semi-supervised learning.

To further build a light-weight semi-supervised model, we also propose a heterogeneous knowledge distillation (HKD) module, which enables the prior knowledge transfer between heterogeneous models. We also propose a novel confidence-guided labeling loss to better train the proposed HKD module.

The contribution of this is summarized as follow:

- A new training paradigm is proposed for supervised singing melody extraction task, which uses harmonic supervision to classify the audio pixels in the spectrogram into four classes. The proposed harmonic supervision method encourages the model to learn positional correlations in the spectrogram, reducing the octave errors.
- Extending from supervised singing melody extraction, we apply harmonic supervision to the semi-supervised scenario as a consistency regularization method to improve the accuracy of pseudo labels.
- To further build a light-weight semi-supervised model, a heterogeneous knowledge distillation module is proposed to enable the prior knowledge can be transferred between heterogeneous models. We also propose a novel confidence-guided labeling loss to better train the proposed HKD module.
- We use MIR-1K dataset and part of music tracks of the MedleyDB dataset as labeled data for training the model and we evaluate the performance on the well-known ADC2004, MIREX 05, iKala and another part of MedleyDB. The experimental results demonstrates the superiority of our method compared with other state-of-the-art ones.

## 2 RELATED WORKS

### 2.1 Supervised Singing Melody Extraction

Deep learning models for the singing melody extraction task undergone various model architectures throughout its history. Kum et al. [31] proposes a multi-column deep neural network to learn a nonlinear mapping between frame and melody. Subsequently, many convolutional neural network (CNN) based approaches have been developed to better capture spectral-temporal information [9, 14, 16, 33, 42]. In addition, the use of musical prior knowledge and structural priors has further broadened the design of melody extraction models [11, 17, 21, 30, 35]. The relationship between frequencies can be further captured through multi-dilation or attention networks [13, 15, 50, 52], or harmonic constant-Q transform (HCQT) [5]. The separate prediction of octave and pitch-class is proposed in [8] to further enhance the octave accuracy and chroma accuracy of the melody extraction. These models further improve the melody extraction performance.

### 2.2 Semi-supervised Singing Melody Extraction

Although semi-supervised learning is a crucial method to handle the unlabeled data in the era of artificial intelligence, merely little works [29, 48, 51] have been studied for singing melody extraction. As far as we know, Kum et al. [29] employed a pretrained teacher model to generate pseudo labels on unlabeled data. However, since there is a lack of the process of data selection, it is prone to generate wrongly predicted pseudo labels, which would decrease the performance of

**Figure 2: The framework of the proposed HKDSME. The proposed framework HKDSME consists of three modules: (a) harmonic supervision (HS) module, (b) harmonic consistency regularization (HCR) module and (c) heterogeneous knowledge distillation (HKD) module.**

the model. Yu et al. [51] proposed a few-shot learning algorithm to address the imbalance distribution of the samples due to the scarce of labeling data. Unfortunately, this algorithm can not be used in the scenario of semi-supervised learning. In the field of machine learning, consistency regularization has become an active research direction [10, 24, 34, 38, 55]. A lot of popular consistency regularization methods [2, 4, 32, 43, 47] have been proposed to deal with the unlabeled data. Unfortunately, Yu et al. [48] has claimed that the singing melody extraction task is too sensitive to obtain satisfied results. In this work, we propose a harmonic supervision based consistency regularization method.

## 2.3 Knowledge Distillation

Our work is related to knowledge distillation, which is initially proposed in [20]. Hinton et al. [20] proposed to use the output class probabilities of a static cumbersome model as soft targets to teach a light-weight student model. Zagoruyko et al. [54] proposed a method that encourage the student model to generate the same attention as the teacher model. Wu et al. [46] proposed a mutual learning method to learn complementary features in semi-supervised learning. However, these methods mentioned above are either logit-based or intermediate-feature based methods, which are too heavy to train the model. In this paper, we propose a confidence guided loss incorporated with HKD module to dynamically adjust the amount of information learned from teacher model.

## 3 METHODOLOGY

The overview of the proposed framework is presented in Fig.2. We choose to use FTANet [50] as the feature extractor of HKDSME. The proposed framework HKDSME consists of three modules: harmonic supervision (HS) module, harmonic consistency regularization (HCR) module and heterogeneous knowledge distillation (HKD) module. At the same time, we perform HS module, HCR module and HKD module between the two groups of the generated music representations. We will introduce each components in the following subsections.

## 3.1 Semi-supervised Learning Setup

In this paper, the inputs are from both labeled and unlabeled data. For the input data, the music signal can be denoted as $D = \{D_l, D_u\}$. $D_l = \{(x_1, y_1), (x_2, y_2), ...(x_m, y_m)\}$ and $D_u = \{u_1, u_2, ..., u_N\}$ denote the labeled music data and unlabeled music data, respectively. $M$ and $N$ are the number of labeled and unlabeled data. $T$ denotes the whole training dataset. The learning objective function is constructed in the following form:

$$\min_{\theta}\{L_l(D_l, \theta) + \omega L_u(D_u, \theta)\}, \quad (1)$$

where $L_l$ is the loss function of the supervised learning and $L_u$ is the loss function of the unsupervised learning. $\omega$ is a non-negative parameter, $\theta$ represents the parameters of our proposed framework.

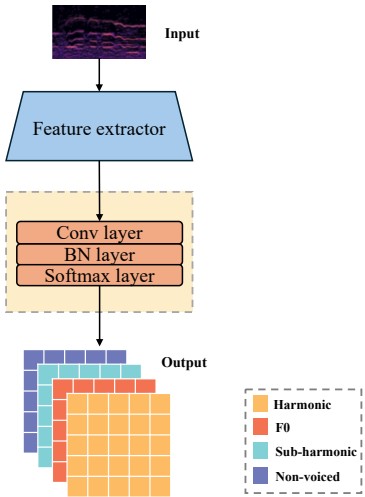

**Figure 3: Illustration of the proposed harmonic supervision module.**

## 3.2 Harmonic Supervision for Singing Melody Extraction

The aim of the harmonic supervision (HS) module is to encourage the model to learn the positional correlation relationship in the spectrogram. Distinct from the existing binary classification paradigm, the HS module force the model to predict not only melodic and non-melodic pixels in the spectrogram, but harmonic and sub-harmonic pixels. To be specific, we employ a convolution layer with the kernel size of $1 \times 1$, and the number of output channel of the convolution layer is set to 4. The convolution layer is followed by the batch normalization and Softmax Layers. The output of the module is four feature maps: 1) a feature map with melodic pixels and non-melodic pixels, 2) a feature map with harmonic pixels and non-melodic pixels, 3) a feature map with sub-harmonic pixels and non-melodic pixels and 4) a feature map with non-voiced pixels and non-melodic pixels as shown in Fig. 3. Note that *non-voiced pixel*, as an auxiliary flag, indicate that there is no singing voice at that time step. To be clear, given an input spectrogram $S$, the output $P$ (the melody contour, harmoic/sub-harmonic contour and non-voice contour) can be predicted as follow:

$$FM = HS(f(s)),$$
$$P = argmax(FM), \tag{2}$$

where $f(\cdot)$ denotes the feature extractor (e.g., FTANet without classification head) for singing melody extraction, $HS(\cdot)$ denotes the proposed harmonic supervision module and $FM = \{FM_i | i \in \{1, 2, 3, 4\}\}$. Since the HS module follows the plug-and-play fashion, it can be used as classification head on all of the existing deep learning based singing melody extraction models.

## 3.3 Harmonic Consistency Regularization for Singing Melody Extraction

The aim of the harmonic consistency regularization (HCR) module is to select the generated pseudo labels based on the essential

---

**Algorithm 1** The detailed procedures of HCR

**Input:** Labeled dataset $D_l = \{(x_1, y_1)...(x_n, y_n)\}$,
    Unlabeled dataset $D_u = \{u_1, ..., u_M\}$
**Output:** Training dataset $D$.
1:  $D = \{D_l\}$
2:  **for** $i \leftarrow 1$ to M **do**
3:     # $T$ denotes the time steps of a sample.
4:     $P = \{(y_t^{\frac{1}{3}f0}, y_t^{\frac{1}{2}f0}, y_t^{f0}, y_t^{2f0}, y_t^{3f0}) | t \in (1, T)\} \leftarrow HS(u_i)$
5:     $Num \leftarrow 0$
6:     **for** $t \leftarrow 1$ to $T$ **do**
7:       **if** $3y_t^{\frac{1}{3}f0} == 2y_t^{\frac{1}{2}f0} == y_t^{f0} == \frac{1}{2}y_t^{2f0} == \frac{1}{3}y_t^{3f0}$ **then**
8:         $Num + +$
9:       **end if**
10:    **end for**
11:   **if** $Num == T$ **then**
12:     add $(u_i, y_t^{f0})$ to D
13:   **else if** $Num == 0$ **then**
14:     discard the sample $u_i$
15:   **else**
16:     $\tau \leftarrow Num/T$ # confidence score
17:     # use the confidence score to decide add or discard to $D$
18:     $action \leftarrow 1$ if random.uniform(0, 1) $< \tau$ else 0
19:     add $(u_i, y_t^{f0})$ to D if action==1 else discard $u_i$
20:   **end if**
21: **end for**
22: **return** $D$

---

characteristics of the musical audio and reduce the computation. Given an unlabeled spectrogram, we first employ HS module to predict the following key values related to the singing melody: f0, 2f0, $\frac{1}{2}$f0, 3f0, $\frac{1}{3}$f0.

After obtaining the values, we can validate the predictions mentioned above. If the predicted values are according to the corresponding ratio, we can add the input into the training data for next iteration of the training. Otherwise, there are two kinds of cases need to be discussed: i) If the values are all wrong, the input data will be discarded directly. ii) If not all of the values are right, the input data we will give a low confidence score to the input data. For example, if there are four out of five values are right, we will give a confidence score of 80% to the input data[1]. The detailed procedures are presented in Alg. 1. By this way, we can judge the availability of the unlabeled data. In addition, since this method does not need perform data augmentation, not only can we reduce the cost of computation by data augmentation, but we can avoid the performance decrease comes from perturbations to the sensitive spectrogram by data augmentation.

## 3.4 Heterogeneous Knowledge Distillation For Singing Melody Extraction

In order to build a light-weight semi-supervised model for singing melody extraction, we propose a heterogeneous knowledge distillation (HKD) module. The detailed HKD module is presented in Fig.4. To achieve this, we first employ a large pre-trained model

---

[1]Obviously, more complex scoring criteria can be explored for the task in future.

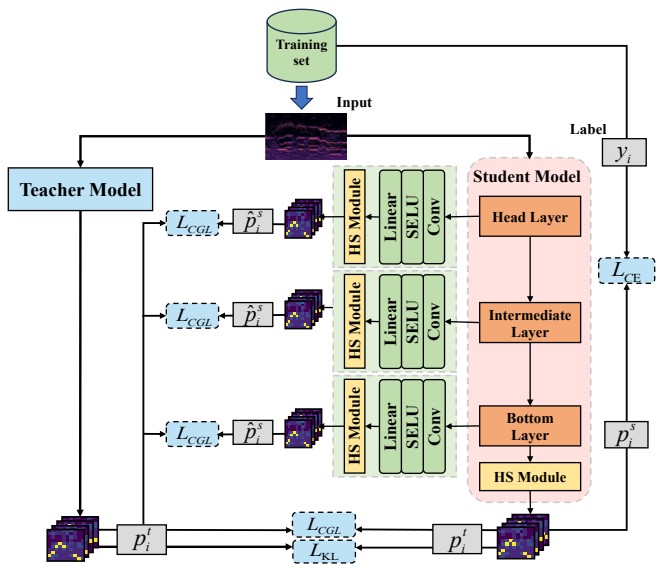

**Figure 4: Illustration of the proposed harmonic consistency regularization module. The parameters are shared for the three-layer small neural network (Conv-SELU-Linear).**

(e.g., FTANet) as the teacher model, and a light-weight model (e.g., MSNet) as the student model. Then we choose several layers from the student model, and we divide the chosen layers into three groups: *head layers, intermediate layers and bottom layers*. Unlike previous methods [19, 25, 36, 45], we employ a three-layer small neural network to transform the feature maps from the chosen layers and the feature map of the last layer from the teacher model into the shared hidden space. Then the information from teacher and student models can be fused and used for training and testing.

Formally, given an input spectrogram $S$, we first feed $s$ into the teacher model and obtain the prediction $p_i^t$. We then feed $s$ into the student model and obtain $p_i^s$. In order to avoid the collapse of the model, we freeze the parameters in the teacher model. After obtaining $p_i^t$ and $p_i^s$, There are three loss functions are calculated. We first perform cross entropy loss function between $p_i^s$ and $y_i$:

$$L_{ce} = \frac{1}{|D|} \sum_i CE(p_i^s, y_i), \qquad (3)$$

where $|D|$ denotes the number of training data and $CE(\cdot)$ is the cross entropy loss function. Then, KL divergence is employed to calculate the difference between predictions from teacher and student model:

$$L_{KL} = \frac{1}{|D|} \sum_i KL(p_i^s, p_i^t), \qquad (4)$$

where $KL(\cdot)$ denotes the KL divergence loss function. By using the two loss functions the outputs among teacher, student and the ground truth could be aligned.

Although we can get a high performance from the teacher model, the teacher model will still generate wrong pseudo labels. To improve the accuracy of the pseudo labels generated by the teacher model, we propose a novel confidence-guided labeling (CGL) loss to achieve this. Since the teacher model will output a probability

**Table 1: The detailed descriptions of the datasets for training and testing the proposed framework HKDSME.**

|  | Dataset | # of Tracks | Duration |
|---|---|---|---|
| Training (Labeled) | MIR-1K | 1000 | 2h 13min |
|  | MedleyDB | 35 | 2h 20min |
| Training (Unlabeled) | FMA | 700 | 5h 15min |
|  | RWC | 30 | 3h 20min |
| Testing | ADC2004 | 12 | 4min |
|  | MIREX 05 | 9 | 4min |
|  | Medley DB | 12 | 48min |
|  | iKala | 262 | 2h 6min |

value in addition to the prediction, the proposed CGL loss is based on how much confidence the teacher model shows. If the prediction from teacher model is not that confident, then the output from student model may stick to its own predictions. Otherwise, the student should obey the prediction from teacher model. The CGL loss can be calculated:

$$L_{CGL} = \frac{-1}{|D|} \sum_i (1 + g_c^t)^\alpha \log g_c^s + \sum_{y \setminus c} g_{\hat{c}}^t \log g_{\hat{c}}^s, \qquad (5)$$

where $c$ denotes the index of frequency bin corresponding to the f0, $\hat{c}$ denotes the remaining frequency bins are not f0, $\alpha$ is a hyper-parameter to scale the term $(1 + g_c^t)$, $g_c^t \in (0, 1)$ denotes the confidence shows by the teacher model and $g_c^s \in (0, 1)$ denotes the confidence shows by the student model. The overall loss function of the proposed HKDSME framework can be calculated:

$$L = L_{CE} + L_{KL} + L_{CGL} \qquad (6)$$

## 4 EXPERIMENTS

### 4.1 Datasets

We train and evaluate our proposed HKDSME framework on several public datasets, the descriptions of the datasets we used are listed in Table 1. For the training data, we first choose 1000 popular music tracks from MIR-1K [22] and 35 popular music tracks from MedleyDB [6] with melody annotated. Then we also choose 700 popular music tracks from FMA dataset [12] without labels. We also use 30 popular music tracks from RWC dataset [18]. For the testing data, we use four well-known testing datasets for this task: 12 tracks from ADC2004, 9 tracks from MIREX05[2], 12 tracks from MedleyDB and 262 tracks from iKala [7].

### 4.2 Experiment Setup

Following the convention in the literature [39], we use the following metrics for performance evaluation: overall accuracy (OA), raw pitch accuracy (RPA), raw chroma accuracy (RCA), voicing recall (VR) and voicing false alarm (VFA). We use mir eval library [37] with the default setting to calculate the metrics. For each metric other than VFA, the higher score, the higher performance. In the literature, OA is often considered more important than other metrics.

---

[2]https://labrosa.ee.columbia.edu/projects/melody

**Table 2: The performances of the proposed HKDSME and baseline methods on the ADC2004 and MIREX 05 datasets, the values in the table are percentile.**

| Dataset Methods | ADC2004 | | | | | MIREX 05 | | | | |
|---|---|---|---|---|---|---|---|---|---|---|
| | OA | RPA | RCA | VR | VFA | OA | RPA | RCA | VR | VFA |
| DSM | 68.1 | 66.5 | 69.1 | 76.3 | 17.4 | 72.1 | 74.3 | 75.4 | 77.2 | 30.1 |
| MSNet | 77.1 | 75.1 | 75.8 | 80.8 | 16.3 | 82.0 | 78.4 | 79.0 | 82.3 | 14.3 |
| MD+MR | 76.5 | 77.7 | 78.1 | 78.4 | 22.8 | 79.0 | 75.6 | 76.7 | 79.9 | 25.4 |
| Teacher-student | 78.5 | 77.9 | 78.4 | 81.5 | 14.1 | 81.7 | 76.3 | 76.9 | 81.8 | 14.8 |
| FTANet | 77.4 | 76.3 | 76.5 | 83.2 | 13.3 | 84.4 | 77.8 | 77.8 | 83.9 | **5.2** |
| HGNet | 75.3 | 74.8 | 75.1 | 80.9 | 21.3 | 82.1 | 75.4 | 76.3 | 80.5 | 21.7 |
| MCSSME | 79.7 | 78.1 | 78.9 | 80.8 | 16.3 | 84.6 | 80.1 | 80.5 | **85.3** | 15.9 |
| **HKDSME** (ours) | **85.6** | **85.2** | **85.3** | **87.5** | **11.5** | **85.7** | **82.3** | **82.4** | 84.1 | 5.4 |

**Table 3: The performances of the proposed HKDSME and baseline methods on the MedleyDB and iKala datasets, the values in the table are percentile.**

| Dataset Methods | MedleyDB | | | | | iKala | | | | |
|---|---|---|---|---|---|---|---|---|---|---|
| | OA | RPA | RCA | VR | VFA | OA | RPA | RCA | VR | VFA |
| DSM | 65.3 | 50.8 | 52.0 | 62.1 | 21.3 | 71.2 | 78.4 | 79.1 | 79.4 | 23.6 |
| MSNet | 67.4 | 52.2 | 52.8 | 54.5 | 12.2 | 78.0 | 80.1 | 81.2 | 80.2 | **13.6** |
| MD+MR | 68.3 | 53.1 | 53.9 | 58.3 | 18.6 | 78.5 | 80.7 | 81.9 | 79.2 | 29.8 |
| Teacher-student | 69.8 | 53.6 | 54.4 | 61.2 | 20.7 | 77.1 | 77.2 | 78.3 | 79.5 | 35.4 |
| FTANet | 70.2 | 54.2 | 55.8 | 60.4 | 15.3 | 80.7 | 80.8 | 81.4 | 83.1 | 23.7 |
| HGNet | 69.3 | 53.3 | 53.8 | 61.3 | 13.4 | 79.3 | 80.4 | 80.8 | 82.5 | 22.8 |
| MCSSME | 71.4 | 56.4 | 57.0 | 63.4 | 16.8 | 81.2 | 81.4 | 81.7 | 84.2 | 21.9 |
| **HKDSME** (Ours) | **72.3** | **60.8** | **62.5** | **66.4** | **12.1** | **82.1** | **83.0** | **83.2** | **85.3** | 14.9 |

The proposed framework is implemented using PyTorch [3]. All experiments are conducted on a machine with two NVIDIA RTX 3090 GPUs. For a fair comparison, we train the baseline models using the same training data. Following [21], we choose to use a set of input representations. It contains three parts: (1) the generalized cepstrum(GC) [28], (2) the generalized cepstrum of spectrum (GCoS) [41], (3) the Combined Frequency and Periodicity (CFP) spectrum [42]. In this work, the audio files are resampled to 8 kHz and merged into one mono channel following [50]. Data representations are computed with a Hanning window of 768 samples and hop size of 80 samples. To adapt the pitch ranges required in singing melody extraction, following [21], we set hyper-parameters in computing the CFP for our model. For vocal melody extraction, the number of frequency bins is set to 320, with 60 bins per octave, and the frequency range is from 31 Hz (B0) to 1250 Hz (D#6).

### 4.3 Comparison with State-of-the-art Methods

We compare our framework with seven state-of-the-art (SOTA) methods for singing melody extraction: (1) DSM [5], (2) MSNet [21], (3) MD+MR [15], (4) Teacher-student [29], (5) FTANet [50], (6) HGNet [49], (7) MCSSME [48]. To demonstrate the effectiveness of our proposed method, we train the proposed framework HKDSME and compare our method with other baseline methods. The quantitative results are shown in Table 2 and Table 3. It is

observed that with assisted unlabeled music data, our proposed HKDSME achieves the best performance on four public testing sets in general. For comparison with other baselines, when focusing on OA, the proposed method outperforms FTANet by 10.6% in ADC2004, by 1.5% in MIREX 05, by 3.0% in Medley DB and by 1.7% in iKala, relatively. When comparing with semi-supervised methods on OA, the proposed method outperforms Teacher-student by 9.0% in ADC2004, by 4.9% in MIREX 05, by 7.6% in Medley DB and by 6.5% in iKala, relatively. It is worthy to mention that the effectiveness of our method is from the harmonic consistency regularization and harmonic knowledge distillation compared with the performance of a semi-supervised model Teacher-student [29].

### 4.4 Case Study

To investigate what types of errors are solved by the proposed model, a case study is performed on several music tracks chosen from ADC2004 and MIREX 05 datasets. We choose FTANet [50] to compare with due to its effectiveness and popularity. As depicted in Fig. 5, we can observe that there are fewer octave errors in our model than in FTANet in general. We can also observe that there are some errors that are wrongly predicted near the right frequency bin in diagram (a) near the time of 1000 ms, which are correctly predicted in diagram (c). Through the visualization of the predicted melody contour, we can say that the performance gains of the proposed model can be attributed to solving the octave errors and other

---

[3]https://pytorch.org

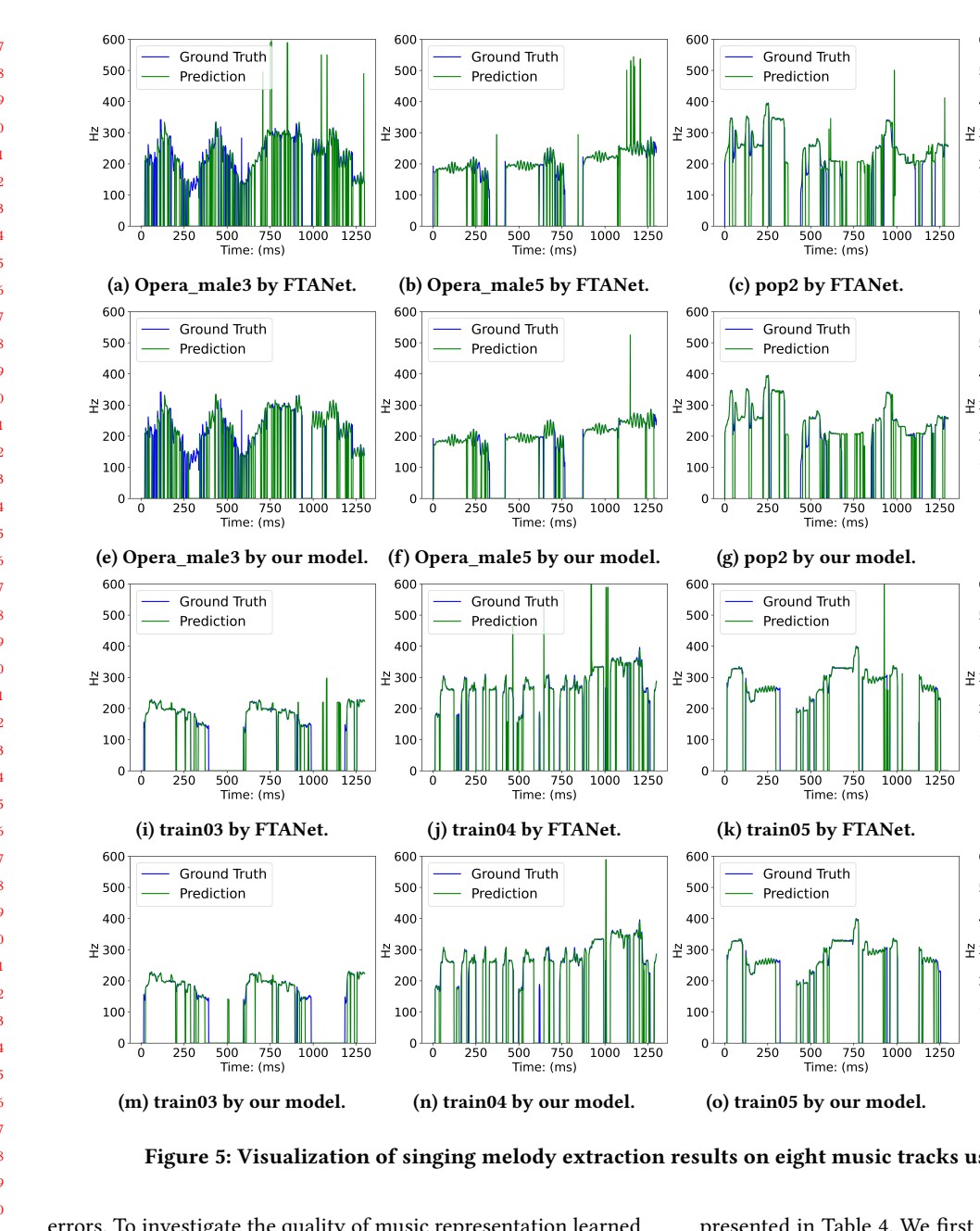

Figure 5: Visualization of singing melody extraction results on eight music tracks using different models.

errors. To investigate the quality of music representation learned from our proposed HKDSME, we visualize the learned representation via t-SNE. We use 12 popular music tracks to perform t-SNE, as observed in Fig. 6. The left is the distribution of binary classification, and the right is the distribution of harmonic supervision. The representations on the right are well clustered. Owing to the proposed HKDSME framework, the predictions of our method have smoother contours and the examples with the same frequency are closer to each other.

## 4.5 Ablation Study

To investigate the effectiveness of the key components in our framework, we conduct ablation studies and the quantitative results are

presented in Table 4. We first remove the harmonic supervision module and use binary classification to train the framework. As observed in Table 4, the performances of OA decreased by 4.7% in ADC2004 and 4.4% in MIREX 05. We then remove the HCR module and use data augmentation to select unlabeled data, the performances of OA decreased by 3.3% in ADC2004 and 3.9% in MIREX 05. The observation indicates that the use of harmonic consistency regularization helps improve the performance of singing melody extraction. Next, we remove the heterogeneous knowledge module and use intermediate features of the teacher and student models for knowledge distillation, the performances of OA decreased by 1.1% in ADC2004 and 1.6% in MIREX 05. Finally, we remove the CGL loss, the performances of OA decreased by 0.7% in ADC2004 and

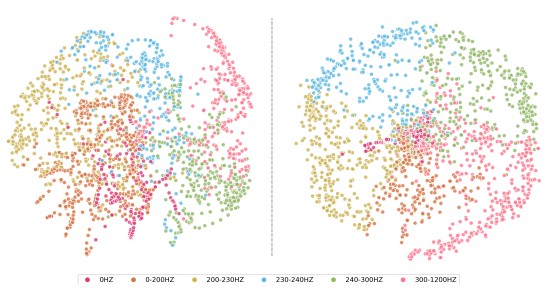

Figure 6: Visualization of the learned music representation via t-SNE. The left is the feature distributions using binary classification, the right is the feature distributions using harmonic supervision. Different colors denote various examples with different frequencies.

Table 4: Results of Ablation Study on ADC2004 and MIREX 05 dataset. The values in the table are percentile. "w/o HS" and "w/o HCR" denote without harmonic supervision and harmonic consistency regularization respectively. "w/o HKD" stands for without heterogeneous knowledge distillation. "w/o CGL" stands for without confidence-guided labeling loss.

| Dataset Methods | ADC2004 | | | MIREX 05 | | |
|---|---|---|---|---|---|---|
| | OA | RPA | RCA | OA | RPA | RCA |
| w/o HS | 81.6 | 81.1 | 81.8 | 81.9 | 78.2 | 79.0 |
| w/o HCR | 82.8 | 83.1 | 83.2 | 82.4 | 78.9 | 79.0 |
| w/o HKD | 84.7 | 84.3 | 84.6 | 84.3 | 80.8 | 80.9 |
| w/o CGL | 85.0 | 84.8 | 84.9 | 85.1 | 81.8 | 81.9 |
| HKDSME | **85.6** | **85.2** | **85.3** | **85.7** | **82.3** | **82.4** |

Table 5: Effects of harmonic supervision.

| Dataset Har. Combinations | ADC2004 | | | MIREX 05 | | |
|---|---|---|---|---|---|---|
| | OA | RPA | RCA | OA | RPA | RCA |
| f0 | 81.6 | 81.1 | 81.8 | 81.9 | 78.2 | 79.0 |
| f0+$\frac{1}{2}$f0+2f0 | 81.9 | 80.3 | 80.5 | 83.4 | 80.0 | 80.1 |
| f0+$\frac{1}{3}$f0+3f0 | 81.8 | 80.2 | 80.3 | 82.7 | 79.6 | 79.8 |
| f0+2f0+3f0 | 83.2 | 82.0 | 82.1 | 83.2 | 80.3 | 80.4 |
| f0+$\frac{1}{2}$f0+$\frac{1}{3}$f0 | 83.2 | 82.0 | 82.1 | 83.5 | 80.1 | 80.2 |
| f0+$\frac{1}{2}$f0+$\frac{1}{3}$f0+2f0+3f0 | **85.6** | **85.2** | **85.3** | **85.7** | **82.3** | **82.4** |

0.7% in MIREX 05. The results show that the proposed harmonic supervision paradigm contributes most to HKDSME. And the proposed HCR and HKD modules can also improve the performances in the scenario of using very limited labeled data. Overall, the key components of our framework HKDSME are tightly incorporated and collaboratively devote to remarkable results.

**Effects of harmonic supervision.** To investigate how the supervision improves the performance, we vary the harmonic numbers of harmonic supervision. The quantitative results is presented

Table 6: Effects of $\alpha$.

| Dataset $\alpha$ Settings | ADC2004 | | | MIREX 05 | | |
|---|---|---|---|---|---|---|
| | OA | RPA | RCA | OA | RPA | RCA |
| $\alpha = 0.5$ | 84.6 | 84.3 | 84.4 | 84.9 | 81.6 | 81.7 |
| $\alpha = 1$ | 84.9 | 84.5 | 84.6 | 85.4 | 82.0 | 82.1 |
| $\alpha = 1.5$ | **85.6** | **85.2** | **85.3** | **85.7** | **82.3** | **82.4** |
| $\alpha = 2$ | 84.3 | 84.0 | 84.1 | 84.5 | 81.1 | 81.2 |
| $\alpha = 2.5$ | 84.2 | 83.9 | 84.0 | 84.5 | 81.1 | 81.2 |

in Table 5. To compare with the binary classification, we also add the results of f0 in the table. Specifically, we perform five combinations of harmonics to justify the effectiveness of our proposed HS module. We first try the combination of f0, nf0 and $\frac{1}{n}$f0, the results show that using the above combination will improve both of the peformances on ADC2004 and MIREX 05. It is worthy to mention that the improvements on MIREX 05 are better than in ADC2004, that is because MIEX05 are popular music tracks, which are prone to be misidentified by sub/harmonic information during extracting f0. Then we try the combination of f0 and harmonics, the results show that on both datasets, the OAs are improved by 2.0% on ADC2004, and by 1.6% on MIREX 05, when compared with binary classification. We also try the combination of f0 and sub-harmonics, the OAs are improved by 2.0% on ADC2004, and by 2.0% on MIREX 05. Although not shown in the table, we tried more sub/harmonics, the results are nearly the same as the best performance in Tab. 5

**Effects of $\alpha$.** We vary the value of $\alpha$ for the CGL loss, and the results is shown in Table 6. To observe the effects of $\alpha$, we vary the $\alpha$ from 0.5 to 2.5. We can observe from Tab. 6, when $\alpha$ is set to 0.5, we can obtain the best performance. We can also observe that the results are nearly the same, which indicates that the proposed framework HKDSME is not very sensitive to $\alpha$.

## 5 CONCLUSION

In this paper, we proposed a heterogeneous knowledge distillation framework for semi-supervised singing melody extraction using harmonic supervision, called HKDSME. Specifically, we proposed a new training paradigm that uses harmonic supervision to classify the audio pixels in the spectrogram into four classes. Extending from harmonic supervision, we proposed to use the extracted harmonics to judge the availability of unlabeled data in terms of inner positional relations. To build a light-weight semi-supervised model, we proposed a heterogeneous knowledge distillation, this module enables the prior knowledge can transfer between different architectures of the models. To further improve the accuracy of pseudo labels, we also proposed a confidence-guided labeling loss function. HKDSME evaluates on a set of well-known public melody extraction datasets with promising performances. The experimental results demonstrate the effectiveness of the HKDSME framework for singing melody extraction from polyphonic music using very limited labeled data scenarios. This work has also provided another verification of the feasibility for integrating harmonic supervision and heterogeneous knowledge distillation for semi-supervised singing melody extraction.

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
