# OpenReview forum: "HKDSME: Heterogeneous Knowledge Distillation for Semi-supervised Singing Melody Extraction Using Harmonic Supervision"
_acmmm.org/ACMMM/2024/Conference — MM2024 Oral_

### Official Review · Reviewer_cAfv · 2024-05-06

**Rating:** 5
**Confidence:** 3

**Summary:**

This paper presents HKDSME, which leverages heterogeneous knowledge distillation (HKD) and harmonic supervision (HS) for semi-supervised singing melody extraction. For the HS, the authors propose a multi-target classification strategy to help deep learning models to tell the singing part from other information in the spectrograms (e.g. harmonics and sub-harmonics). Then, the authors propose a harmonic consistency regularisation (HSR) module to select the predicted labels by HS and achieve higher-quality data augmentation accordingly. Finally, in the singing melody extraction part, HKD is utilised to build a light-weight melody extractor (student model) with the reference to a large pretrained model (teacher model). Experimental results and ablation studies advocate the effectiveness of the proposed HKDSME framework, as well as the usefulness of each component in the framework.

**Strengths:**

* This paper is clearly-written and easy to follow.
* The methodology is well motivated and elaborately designed. Using multiple targets and HKD is an intuitive and clever idea to address existing problems.

**Limitations:**

* Line 686: Instead of using "several music tracks", report the actual number of tracks, name & genre of tracks etc. (e.g. are you using the same genres) to highlight the objectivity of your track selection. The same goes for Line 742.
* Section 4.4: instead of relying solely on figure 5, some real statistics should be reported (e.g. percentages of each error, p values from statistical comparisons between groups) to support the advantage of the proposed method over other baselines.
* In addition to the evaluation metrics, some **demonstrations** (including the extracted melodies from HKDSME and other baselines) are recommended to help the reviewers understand the contribution and effectiveness of this work. Currently there are not any in the paper or the supplementary materials.
* Some minor nitpicks that may need double checks:
	* Line 177: enables the prior knowledge **to** transfer...
	* Line 335: Should $m$ in $(x_m, y_m)$ in $D_l$ be capitalised as $(x_M, y_M)$? In line 336 the authors denote $M$ as the total number of labelled data.
	* In footnote 1, "Obviously" does not appear academically appropriate.
	* In Line 499, does the lower case $s$ refer to the input spectrogram? (if that's true, should it be capitalised?)
	* Line 547: I think it should be "$\hat{c}$ denotes the remaining frequency bins **that** are not $\texttt{f0}$?".
	* Line 576: "In the literature...", I would recommend adding some citations here and briefly introducing why OA is the most important.

**Suitability:**

2

---

### Official Review · Reviewer_1BP1 · 2024-05-24

**Rating:** 5
**Confidence:** 2

**Summary:**

This paper proposed HKDSME, a semi-supervised framework using heterogeneous knowledge distillation and harmonic supervision for singing melody extraction. Experiments show that HKDSME effectively extracts melodies from polyphonic music with minimal labeled data.

**Strengths:**

1. The novelty of this approach lies in utilizing harmonic, sub-harmonic, and fundamental frequencies (F0), referred to as Harmonic Supervision, to classify pixels on the input spectrogram.
2. The authors proposed a Heterogeneous Knowledge Distillation method, which involves selecting layers from the student model and projecting the outputs according to a specially designed loss function.
3. The authors introduced Harmonic Consistency Regularization, which carefully filters pseudo-labeled data based on the physical relationships of F0, 2F0, 3F0, 1/2F0, and 1/3F0.

**Limitations:**

Please note the following typos:
1. On line 335, the last data pair subscript should be an uppercase 'M'.
2. In Algorithm 1, the labeled dataset is denoted with 'n', while the unlabeled dataset is denoted with 'M'. This notation is inconsistent with line 337.

**Suitability:**

2

---

### Official Review · Reviewer_ivqx · 2024-05-24

**Rating:** 6
**Confidence:** 4

**Summary:**

This paper proposed a singing pitch extraction model for polyphonic music. Specifically, a new training objective is adopted that adds harmonic and subharmonic into classification. A new regularization is proposed to exploit inter-label relationships. Further, this objective is extended in a semi-supervised setting. Knowledge distillation is also adopted for heterogeneous models, with proposed confidence guided labeling loss, to make the model lightweight. Experiments show superior results than SOTA pitch tracking methods. Visualization of the learned representation shows a high correlation with frequency of melody.

**Strengths:**

- The author made important observations that pixels from spectrograms have different importance, and designed new training objectives to add harmonic and subharmonic information in classification. Further, a regulation objective is designed for pitch and its harmonic, which is very intuitive in singing voice where harmonics is a very common scenario.

- A knowledge distillation approach is proposed with a combination of cross entropy loss, KL divergence, and a group of task-related soft labels. Task-related loss is computed for multiple depth of the student models’ output.

- The visualization of features shows the learned feature is highly correlated with melody’s frequency. It indicates the proposed approach could be extended to various pitch-related audio analysis tasks.

**Limitations:**

**Major Issues**
- In Section 3.1, L_l and L_u are not explained. Not sure how they are computed.

**Minor Issues**
- From Table 5, the student model (HKDSME) has better performance than the teacher model (w/o HKD). The reviewer believes this is not a common scenario, so it can be better to include further comparison and discussion of behavior of teacher model and student model.
- TONet [8] seems not included in the comparison (although the proposed model has better performance than that of TONet).

**Suitability:**

3

---

### Official Review · Reviewer_AJzc · 2024-05-24

**Rating:** 4
**Confidence:** 3

**Summary:**

The paper introduces a heterogeneous knowledge distillation framework, HKDSME, for semi-supervised singing melody extraction using harmonic supervision. It adopts a novel training paradigm to classify the audio pixels in the spectrogram into four classes, as well as a light-weight semi-supervised model. Experiment results demonstrate that HKDSME outperforms previous works for singing melody extraction with limited labeled data.

**Strengths:**

1. The proposed approach is well motivated. Singing melody extraction from polyphonic music is a challenging task. Since previous studies treated all pixels in the spectrogram equally and ignored the accuracy of pseudo labels generated by extraction models, HKDSME appears as a logic and promising solution.
2. The paper is well structured and well written.
3. The proposed harmonic supervision module follows the plug-and-play fashion and is applicable to all singing melody extraction models.
4. The heterogeneous knowledge distillation module is lightweight and can make full use of limited labeled data.

**Limitations:**

Editing and Typesetting: Check the paper for spelling mistakes, punctuation usage, word collocations, and other errors to ensure the writing is smooth and refined. Some examples (non-exhaustive list):
- line 32-33: "… which **enables the prior knowledge transfers** between heterogeneous models …"
- line 104-106: "… They are distributed above and below the singing melody line according to a fixed ratio, and **synchronize** with the changes of the singing melody …"
- line 200-203: "… The experimental **results demonstrates** the superiority of our method compared with other state-of-the-art ones …"
- line 282: "… a method that **encourage** the student model …"
- line 376: "… the HS module **force** the model to …"
- line 387: "… as an auxiliary flag, **indicate** that there is no singing voice …"
- ...
Lack of Generated Results: Adding the results generated by different models in the supplementary materials can make the conclusions more convincing.

**Suitability:**

2

---

### Meta-Review · Area_Chair_wDHZ · 2024-06-23

**Recommendation:** Accept (Oral)
**Confidence:** 4

**Metareview:**

In general, this is a well-motivated and well-written paper of good quality.

+ Based on important observations that pixels from spectrograms have different importance, the authors designed new training objectives to add harmonic and subharmonic information, referred to as Harmonic Supervision, in classifying pixels on the input spectrogram.
+ The authors proposed a Heterogeneous Knowledge Distillation method, which involves selecting layers from the student model and projecting the outputs according to a specially designed loss function.
+ The authors introduced Harmonic Consistency Regularization, which carefully filters pseudo-labeled data based on the physical relationships of F0, 2F0, 3F0, 1/2F0, and 1/3F0.

- Numerous typos need to be fixed before publication.